# Ultra-precise quantification of mRNA targets across a broad dynamic range with nanoreactor beads

Ivan Francisco Loncarevic[1]*, Susanne Toepfer[1], Stephan Hubold[1], Susanne Klingner[1], Lea Kanitz[1], Thomas Ellinger[1], Katrin Steinmetzer[1], Thomas Ernst[2], Andreas Hochhaus[2], Eugen Ermantraut[1]

1 BLINK AG, Jena, Germany, 2 Universitätsklinikum Jena, Klinik für Innere Medizin II, Abteilung Hämatologie und Internistische Onkologie, Jena, Germany

* ivan@blink-dx.com

**Data Availability Statement:** All relevant data are within the manuscript and its Supporting information files.

## Abstract

Precise quantification of molecular targets in a biological sample across a wide dynamic range is a key requirement in many diagnostic procedures, such as monitoring response to therapy or detection of measurable residual disease. State of the art digital PCR assays provide for a dynamic range of four orders of magnitude. However digital assays are complex and require sophisticated microfluidic tools. Here we present an assay format that enables ultra-precise quantification of RNA targets in a single measurement across a dynamic range of more than six orders of magnitude. The approach is based on hydrogel beads that provide for microfluidic free compartmentalization of the sample as they are used as nanoreactors for reverse transcription, PCR amplification and combined real time and digital detection of gene transcripts. We have applied these nanoreactor beads for establishing an assay for the detection and quantification of BCR-ABL1 fusion transcripts. The assay has been characterized for its precision and linear dynamic range. A comparison of the new method against conventional real time RT-PCR analysis (reference method) with clinical samples from patients with chronic myeloid leukemia (CML) revealed excellent concordance with Pearsons correlation coefficient of 0.983 and slope of 1.08.

## Introduction

Digital amplification techniques [1] such a droplet digital PCR [2] or PCR on nanofluidic chips [3] offer the advantage of absolute quantification with exquisite precision [4]. Precision, and as such the measurement range of digital assays is a direct function of the number of individual reaction compartments (partitions) used for the analysis [5]. The more extensive the dynamic range the more partitions must be used. However microfluidic tools are needed within the evaluation to split a sample into sub-nanoliter droplets or micro structured substrates for distributing the sample across nano-chamber arrays. Endpoint fluorescence generated by the amplification reaction in each respective partition is used to detect partitions containing amplification targets. Recently, we [6] and others [7, 8] have introduced techniques

**Funding:** The authors' affiliation BLINK AG received funding for this study in the form of grants from The German Federal Ministry of Education and Research (BMBF) (Grant No. 031B0526A) and from Free State of Thuringia and European Regional Development Fund (Grant No. EU 2017 FE 9005). BLINK AG used these grants to provide support in the form of salaries to TE, EE, SH, IL, ST, and SK. The specific roles of these authors are articulated in the 'author contributions' section. The funders had no role in study design, data collection and analysis, decision to publish, or preparation of the manuscript.

**Competing interests:** The authors have read the journal's policy and have the following competing interests: TE, EE, SH, IL, ST, SH, and SK are paid employees of BLINK AG. There are no patents, products in development or marketed products associated with this research to declare. This does not alter our adherence to PLOS ONE policies on sharing data and materials.

for generating digital compartments with hydrogel beads serving as volume templates for aqueous partitions of the sample and thus allowing for microfluidics-free digital assays. Whereas in conventional droplet PCR microfluidic chips are required to partition the sample into thousands of individual droplets here pre-made hydrogel beads provide for the reaction volume for reverse transcription and PCR amplification. A schematic of the developed workflow is shown in Fig 1.

Previously crosslinked polymer beads have been used for templating digital reaction compartments. This approach provided for a small reaction space at the interface between the hydrogel bead and the surrounding oil. In this work we are employing monodisperse non-crosslinked hydrogel beads comprised of agarose and gelatin. The composition has been found mechanically stable to sustain the handling during the assay procedure. Moreover, the matrix binds RNA template in RT-PCR buffer. Another important feature of the bead matrix is its thermal hysteresis with a gelling temperature of 40°C and a melting temperature of 80°C in the employed RT-PCR buffer. This allows for robust handling below the melting temperature and for minimized inhibitory effects at the temperatures used during thermocycling for PCR which would be otherwise observed due to steric hindrance by the gel matrix. Fluorescence detection has been found equally possible at any temperature in hydrogel or liquid state (S1 Fig).

The nanoreactor beads are provided in RT-PCR buffer and are exposed to extracted RNA mixed with RT-PCR reagents. The hydrogel composition is designed to allow the reagents to diffuse into the interior of the beads. Following an incubation step for reagent take-up and nucleic acid binding the beads are transferred to a fluorocarbon oil containing an emulsifier. The bead suspension is then transferred to a reaction and detection chamber (RDC) (Fig 2 and

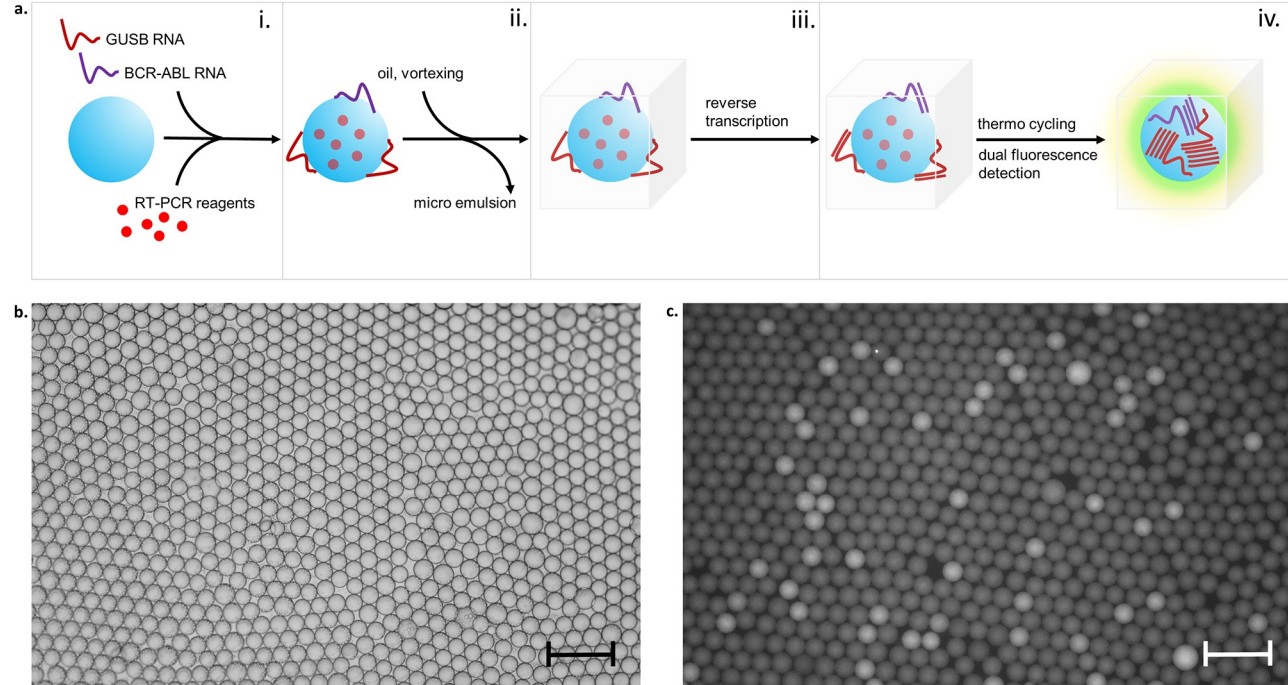

**Fig 1. Nanoreactor bead workflow.** a, Process steps: i.) nanoreactor bead loading with a solution containing amplification reagents and the extracted nucleic acid, ii.) bead transfer to non-aqueous liquid with a suitable emulsifier, iii.) cDNA synthesis on encapsulated hydrogel bead, iv.) bead melting, PCR and fluorescence detection; b, bright field image of bead nanoreactors in PicoSurf oil; c, fluorescence image of nanoreactor beads post PCR; scalebar = 400μm; average diameter of beads was 100μm, average volume 0.5nL.

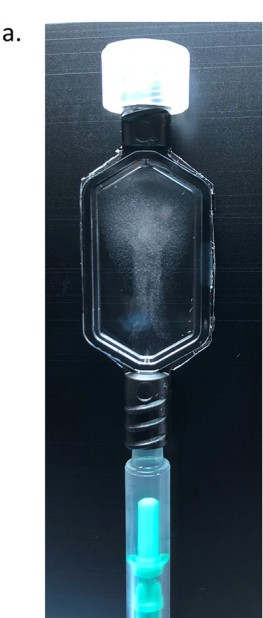
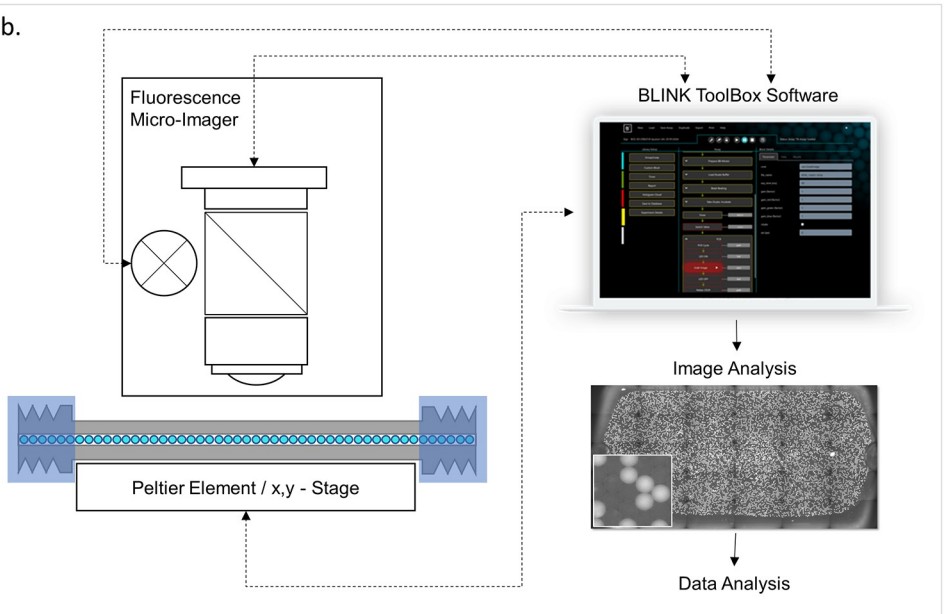

**Fig 2. Incubation and detection set-up.** a, Reaction and detection chamber (RDC) with nominal depth of 100μm and nominal volume of 37,5μL, in the process of bead loading the bead suspension is visible in the RDC; b, schematic of the detection set-up comprising the incubation chamber mounted on a Peltier-element for thermocycling placed on a x,y-stage under an epi-fluorescence microscope; the process is automated through the Blink ToolBox Software package (https://www.blink-dx.com/technology/toolbox).

S2 Fig) designed to conveniently generate a single bead layer for detection of the fluorescence within the individual partitions created by the beads. The RDC features are a luer-lock inlet and a luer-lock outlet on a polycarbonate frame with an attached polycarbonate sheet with a thickness of 100μm and transparent cover part on the opposite side of the frame. The assembly forms a flat chamber between the polycarbonate sheet and the transparent cover with a height of approximately 100μm. The RDC is filled with the bead emulsion and clamped on a temperature control unit whereby the thin polycarbonate layer is brought into close contact with the surface of a Peltier element. The Peltier element is part of a custom-made temperature control unit that is designed to fit on a standard microtiter plate interface on an x,y -stage of an inverted epi-fluorescence microscope equipped with fluorescent filter sets for FAM, Cy3 and Cy5 dyes and with a CMOS camera for fluorescence imaging.

The set-up has been applied to establish a detection format that combines endpoint (digital) and real time detection in each individual sample partition provided by the respective nanoreactor bead. We found that by analyzing only a modest number of 10,000 beads if compared to other digital assay methods an assay with an unprecedented dynamic range can be realized [5, 9–11]. The method was applied to develop an assay for the quantification of the amount of BCR-ABL1 mRNA relative to the internal reference gene transcript GUSB in Chronic Myeloid Leukemia (CML) [12]. CML is a myeloproliferative neoplasm characterized by the presence of the BCR-ABL1 fusion gene [13]. Quantification of the BCR-ABL1 transcript level reflects leukemic burden. Molecular response to treatment is determined based on the ratios of BCR-ABL1 and control gene transcript levels. Here we present the results of the evaluation of the new assay with nanoreactor beads and compare the data with results obtained with the current clinical gold standard.

## Materials and methods

### Nanoreactor beads

A solution containing 1.5% Acetone-insoluble gelatin from bovine skin type A G1890 (Sigma) or porcine skin type B G9391 (Sigma) and 0.5% Low-gelling 2-Hydroxyethyl agarose (A4018, Sigma) has been prepared in nuclease-free water (Carl Roth) and incubated at 50˚C under gentle agitation (750rpm). The solution has been used to generate low dispersity microbeads on the μEncapsulator system (Dolomite microfluidics) using a fluorophobic droplet junction chip (100μm) with a 4-way linear connector to interface the fluidic connection between tubing and chip. Two Mitos P-Pumps were used to deliver the hydrogel solution and the carrier oil (Picosurf-1, Spherefluidics). The system was modified with an integrated heating rig which is placed on top of a hot plate allowing maintaining the gelatin/agarose hybrid solution in liquid state and heating up the driving fluid ensuring consistent temperature for all components and liquids. Picosurf-1 and the hybrid hydrogel solution were both pre- filtered with a 0.22μm filter before they were placed into their respective reservoirs. Temperature of the heating rig is set to 55˚C to heat the gelatin-agarose hydrogel as well as the droplet junction chip. The nanoreactor beads were collected in a tube on ice and incubated for a minimum of 48h at 4–8˚C before phase conversion. Thereafter, the excess of Picosurf-1 was removed from the nanoreactor bead emulsion by aspiration with a pipet and 0.5 volume 1H,1H,2H,2H-perfluorooctanol (PFO; Sigma) and 3 volume $H_2O$ were added to the same volume. The tube was vortexed to break the emulsion and centrifuged for 5s at 2,500 x g. The aqueous phase with the beads was transferred to a fresh tube and the beads were washed 3x with 7 volume $H_2O$. Excess PFO was removed by transferring again the aqueous phase with the beads into a new tube. The nanoreactor beads were washed 4x with an equal volume of nuclease-free water, aliquoted and stored at 4˚-8˚C for further use.

### Samples

Enrollment of patients in this study was approved by the Jena University Hospital (JUH) ethics committee (2719-12/09). Written informed consent was obtained from all individual participants included in the study. There was no specific participant recruitment for the samples in this study. The samples derived from follow up analyses in August 2019 at the molecular-oncology laboratory at Jena University Hospital (JUH). For the comparison analysis, 28 RNA samples were selected with undetectable, medium or high levels of typical BCR-ABL1 transcripts (b2a2 and/or b3a2) (S1 Table). Exclusion criteria were an atypical BCR-ABL1 transcript and an unknown BCR-ABL1 status. RNA was extracted from peripheral blood and from cell lines at using the TRIzol standard procedure at the JUH [14, 15]. The total RNA concentration of the patient samples varied between 62–262 ng/μl. Each RNA stock was divided in two equal parts representing a reference sample and a sample to be processed on the newly developed nanoreactor beads at the Blink AG. The reference sample was processed at the JUH. RT-PCR was performed as a two-step real time rtPCR with SuperScript™ IV VILO™ from ThermoFisher, followed by a PCR as described elsewhere [15]. This assay protocol is referred to as the reference method in this article. The nanoreactor bead sample was subjected to a one-step RT-PCR process at Blink AG. The BCR-ABL1 negative cell line HL60 [16] and the BCR-ABL1 positive cell line K562 [17] have been used to obtain purified RNA each at a concentration of 700ng/μL for generating contrived samples with desired BCR-ABL1/GUSB ratios for the precision analysis.

The BCR-ABL1 and GUSB working standards were quantified based on droplets and nanoreactor beads and are traceable to the calibrant ERM-AD623, a certified reference

**Table 1. List of primers and probes.**

| BCR-ABL1 | forward primer | TCC GCT GAC CAT CAA YAA GGA |
| | reverse primer | CAC TCA GAC CCT GAG GCT CAA |
| | probe | CCC TTC AGC GGC CAG TAG CAT CTG A |
| GUSB | forward primer | GAA AAT ACG TGG TTG GAG AGC TCA TT |
| | reverse primer | CCG AGT GAA GAT CCC CTT TTT A |
| | probe | CCA GCA CTC TCG TCG GTG ACT GTT CA |

material (Plasmid with BCR-ABL1 and GSUB gene sequences; Merck) for quantification of BCR-ABL1 and GUSB DNA.

## Primers and probes

Primers and probes, each at 0.4μM, were used as previously described (ß-Glucoronidase (GUSB), ABL1 [18], and BCR [19]) and shown in Table 1. No additional oligonucleotides were used for reverse transcription on the nanoreactor beads.

## RT-PCR on nanoreactor beads

Nanoreactor beads were loaded with a solution containing target RNA and the RT-PCR reagents by mixing 40μl of sedimented beads with 10μl RNA sample and 50μl 2x RT-PCR solution for 8 min at 20°C at 1000rpm (100μl final volume). The concentration of the resulting 1x RT-PCR Mix was 20mM Tris HCl, 22mM KCl, 22mM NH4Cl, 3mM MgCl2, 0.4 U/μl Hot Start Taq DNA Polymerase (biotechrabbit GmbH), 0.4 mM dNTPs (biotechrabbit GmbH), 1x RT-Mix (biotechrabbit GmbH), 0.1% (w/v) BSA (Sigma), and 0.4μM TaqMan probes and primers for both GUSB and BCR-ABL1 amplifications. Thereafter beads were sedimented at 300g for 30 seconds and the supernatant was removed. Picosurf-1 (100μl) was added to the sediment (30–40μl) and an emulsion was produced by simply sliding the tube over the holes of a microcentrifuge tube rack 20 times at a frequency of approximately 20/s while pressing the tube against the rack surface or by shaking for 2x 5 seconds at level 3 in the Minilys Homogenizer (Bertin Technologies). Microemulsion resulting from the excess of aqueous liquid forming a layer below the beads was removed by aspiration with a pipet. This step was repeated after resuspending the beads and the microemulsion in another 100μl Picosurf-1. The bead suspension was transferred to the RDC with a syringe and the RDC is placed on a PELTIER element (Quick-Ohm, Küpper & Co. GmbH, #QC-71-1.4–3.7M) in a customized test rig. The test rig is mounted on a motorized x,y-stage of an Axio Observer epi-fluorescence Microscope (Zeiss). Using the ToolBox Software (https://www.blink-dx.com/technology/toolbox; a copy of the software is available upon request) the parameters for thermocycling and imaging were programmed and the microscope, Peltier-Element and x,y-stage controlled. Reverse transcription was conducted for 10min at 50°C, followed by PCR (3min 95°C for initial denaturation and 40 cycles of 5sec 95°C, 5sec 60°C for amplification). Images were taken using a 5x objective (field of view 416mm x 2.774mm), pE-4000 (CoolLED Ltd.) light source and a CMOS camera (UI-3260CP-M-GL, IDS). Fluorochrome specific filters for FAM, Cy5 and Rox or Atto550 F36-501, F36-523 and F36-560 (Semrock) were used for imaging in the respective fluorescence channel. Image acquisition was performed after each PCR cycle at 60°C at one discrete position of the RDC. At the end of the PCR run, the entire RDC was scanned at 20°C resulting in 40 individual image positions (end point; digital analysis). Total analysis time from sample loading onto the nanoreactor beads to quantitative results (cp/bead) was 50 minutes. Fig 3 shows exemplary fluorescence images of the GUSB and BCR-ABL1 RNA

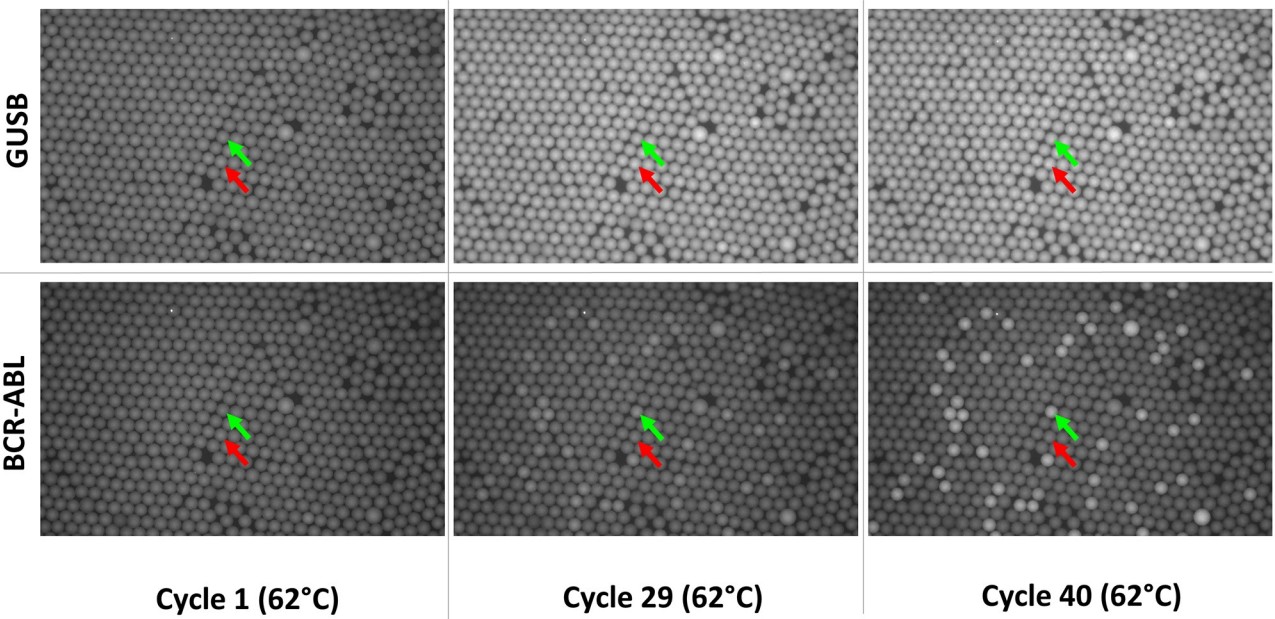

**Fig 3. Fluorescence Images from real time PCR analysis.** Fluorescence images after one, twenty-nine and forty cycles for the GUSB and BCR-ABL1 specific channels (FAM for GUSB, upper row; Cy5 for BCR-ABL1, lower row). Green arrows, one arbitrarily selected BCR-ABL1 positive and GUSB positive nanoreactor bead; Red arrows, a nanoreactor bead negative for BCR-ABL1 and positive for GUSB.

amplification after one, twenty-nine and forty PCR cycles at 62˚C. The images derive from the precision analysis and a sample with 800 copies BCR-ABL1 and 900,000 copies GUSB per reaction (medium level as described in Results). Positive Beads approaching saturation are clearly visible for GUSB after 29 cycles, whereas beads containing single BCR-ABL1 targets show up weakly positive after 29 cycles. The last column shows the corresponding image at 20˚C obtained by scanning of the whole RDC at the end of the run. Discrimination of positive and negative Beads works at both 20˚C and 62˚C while the contrast is best at 20˚C.

## Data analysis

A segmentation algorithm [20] was used to identify the bright disk-shaped nanoreactor beads against dark background in fluorescence images (Fig 3). Circles are fitted to the identified bead contours. Grey value representative for signal intensity and size are determined for each bead. Each bead position is tracked in consecutive images for individual real time PCR analysis.

Target numbers are quantified using either digital PCR analysis based on Poisson statistics or real-time quantitative PCR analysis by comparison of cycle threshold (Ct) value against a calibration curve. The appropriate analysis method is determined based on the number of negative beads in relation to the total number of beads identified in the RDC. For digital PCR, a fluorescence signal threshold is set to distinguish positive beads from negative beads. For real-time quantitative PCR, Ct values are derived from a nonlinear model fitted to the time course of the fluorescence signal for each individual bead. The nonlinear model combines a sigmoid and a linear function where the sigmoid component reveals amplification kinetics, and the linear component represents baseline of the signal. The mean Ct of all beads is a measure for target present in the sample. As a rule, we are applying absolute quantification of Poisson analysis if the proportion of beads remaining negative (and thus dark) after amplification is larger than or equal to 0.5%. Based on a Poisson distribution of targets across beads, a ratio of 0.5%

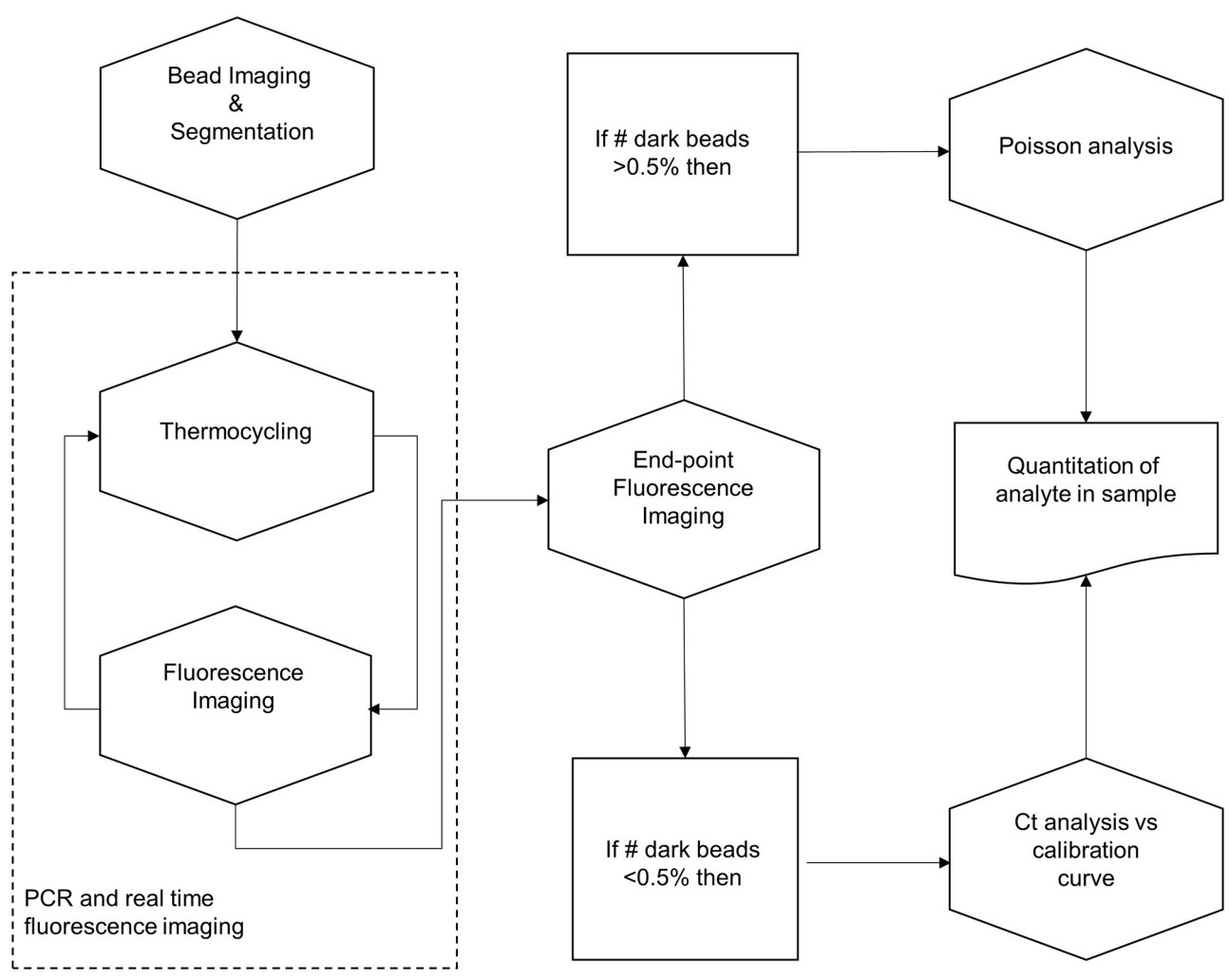

**Fig 4. Analysis algorithm for combined digital and real time analysis.**

negative beads corresponds to an average of 5.3 copies per bead. The resulting upper limit of the digital measuring range is 5.3*N, where N is the total number of beads. If the proportion of negative beads in the imaged area is smaller than this limit of 0.5%, real-time quantitative PCR is utilized. In that case the calibration curve is required.

Digital analysis is based on a Poisson correction to account for the fact that positive beads can contain more than one target. Bead volume variations are factored into the quantitation by performing bead volume specific Poisson corrections. Quantitation for each target is expressed in copies per bead. Real-time quantitative analysis calculates the target number per bead based on the mean Ct value using analyte specific calibration curves. The overall analysis procedure is summarized in Fig 4.

## Statistical analysis

Statistical calculations were performed using R statistical software, version 3.6.1. Analyses for calibration curves and measuring range were conducted for BCR-ABL1 and GUSB results expressed in copies per bead on a logarithmic scale (log10 cp/bead). For both analyses, linear regression models were estimated using ordinary least-squares method. Calibration curve

parameters describe the correlation between mean Ct values from the beads and input copies per bead on a logarithmic scale with formula

$$ct = Offset + Slope \; \cdot log10\frac{cp}{bead}$$

for both targets. The linear model established in the measuring range analysis describes the correlation between measured and nominal BCR-ABL1 copies per bead on a logarithmic scale. For precision and method comparison analyses, also BCR-ABL1/GUSB ratios were evaluated. We used the following formulas to report ratios:

$$\% \; ratio = \frac{BCR - ABL1 \;\; cp/bead}{GUSB \;\; cp/bead} \; \cdot 100\%$$

$$log10\% \; ratio = log10(\% \; ratio)$$

$$MR = log10(100\%) - \; log10(\% \; ratio) = 2 - log10\% \; ratio.$$

The molecular response (MR) is defined as the log reduction level of BCR-ABL1 compared to a control gene under consideration of the conversion factor to ensure traceability to an International Standard (IS). The molecular response was calculated without using the international scale, because a laboratory-specific conversion factor has not yet been established for the new method. Precision was expressed as standard deviation (SD) for each BCR-ABL1/GUSB level separately. For method comparison analysis, the differences between the new proposed method and a reference method were visualized using Bland Altman plot. Deming regression was applied to evaluate the correlation between both methods.

## Results and discussion

We sought to establish an assay for the quantification of transcripts employing nanoreactor beads and characterized the assay for its precision and dynamic range. The proposed data analysis approach allows to extend the limited measuring range of the digital PCR by real-time quantitative PCR analysis for samples with high target numbers. We limited the measuring range of the digital PCR to a proportion of 0.5% negative beads. For a total number of 10,000 nanoreactor beads this corresponds to a minimum of 50 negative beads.

Fig 5 shows the theoretical 95% confidence interval (CI) of the target estimate of digital PCR based on sampling variation and Poisson distribution of targets across the beads (black dashed lines). The estimate would be extremely unreliable for target concentrations above 1 log10 cp/bead (= 10 cp/bead). The introduced upper limit of measuring range of digital PCR with 0.72 log10 cp/bead (= 5.3 cp/bead) (black dotted line) ensures small confidence intervals.

In our approach, application of real-time quantitative PCR is based on the mean Ct value calculated from all beads showing amplification curve kinetics. These are the beads with at least one target molecule. The Ct value being associated with one target per bead is the maximum Ct value that can result from our method. With an increasing number of targets per bead the estimate of the target concentration approaches the real target concentration (blue line). From a target concentration of approximately 0.48 log10 cp/bead (= 3.0 cp/bead) upwards the bias is negligible. Thus, a target concentration of 3 cp/bead can be interpreted as lower limit of measuring range of real-time quantitative PCR (blue dotted line). Based on these considerations, there is on overlap of the measuring ranges of both methods between 3.0 cp/bead and 5.3 cp/bead. Due to the superior precision of digital PCR, we use the digital PCR measuring range to its full extend (Fig 5).

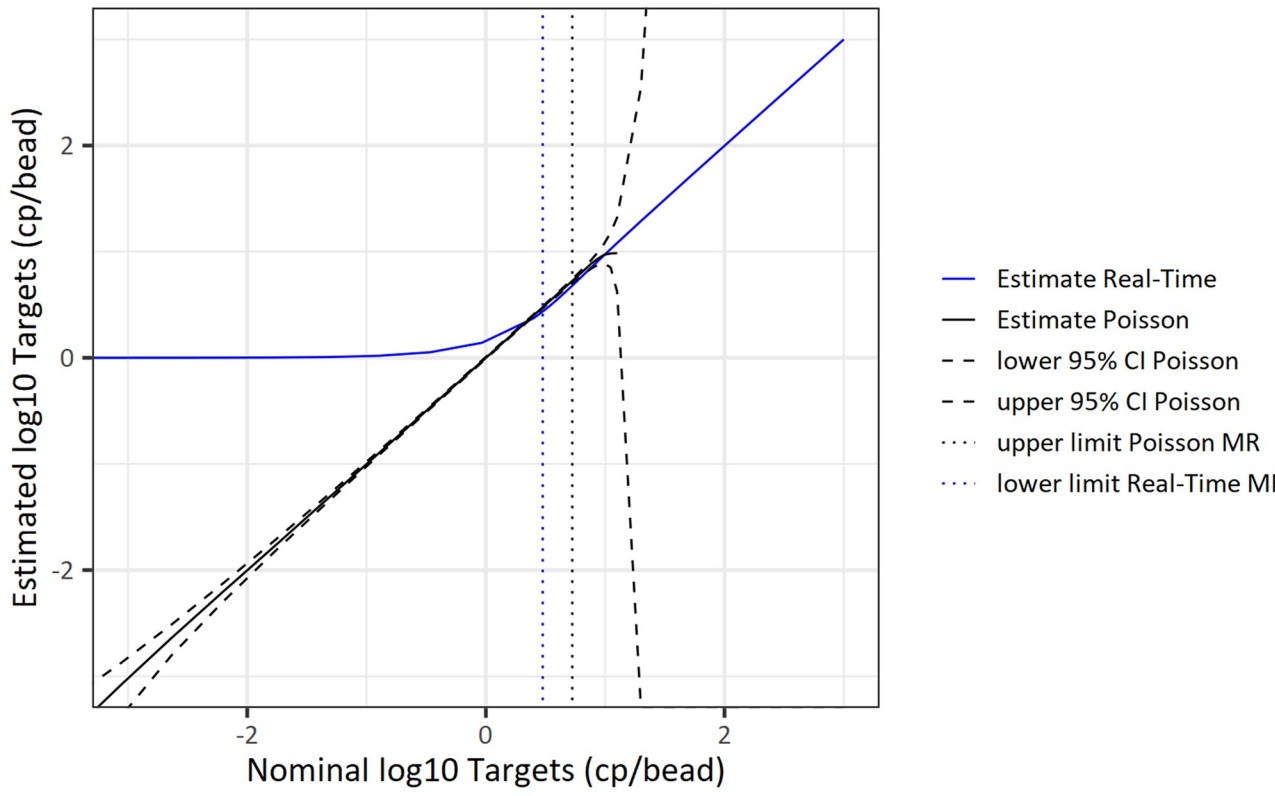

**Fig 5. Overlap of measuring ranges (MR) of digital PCR and real-time quantitative PCR.**

We employed the nanoreactor bead technology for the quantification of BCR-ABL1 and GUSB transcripts and characterized the assay for its precision and dynamic range. We also performed a method comparison against the current clinical standard with clinical samples from patients with CML [21]. The data analysis algorithm requires calibration curves for quantification via real-time quantitative PCR analysis. For this purpose, RNA dilution series with medium and high BCR-ABL1 and GUSB copy numbers were tested independently. The number of copies per bead ranged from 4 cp/bead to 118 cp/bead for BCR-ABL1 and from 7 cp/bead to 251 cp/bead for GUSB. As a result of linear regression, the efficiencies for BCR-ABL1 and GUSB standard curves are 106.5% (slope -3.176) and 100.6% (slope -3.307), respectively. Offsets are 23.459 for BCR-ABL1 and 24.326 for GUSB. Coefficients of determination ($R^2$) are 0.9596 for BCR-ABL1 and 0.9761 for GUSB. In order to assess the measuring range of BCR-ABL1 we tested the assay with 33 different contrived samples with a fixed GUSB concentration and titrated BCR-ABL1 concentration ranging from -3.44 log10 cp/bead (0.00036 cp/bead) to 2.83 log10 cp/bead (676 cp/bead). Quantification results are shown in Fig 6.

The slope for the least-squares regression was determined at 0.97 with 95% confidence interval from 0.95 to 1.00. The estimated intercept was 0.00 with 95% CI from -0.03 to 0.03. Coefficient of determination ($R^2$) was 0.9965 with Pearsons' r of 0.998. As symbols indicate, results were calculated using either Poisson analysis (black dots) from end point fluorescence imaging or Ct-value analysis from real-time fluorescence imaging (blue dots). The data shows linearity and a high degree of concordance between input and measured BCR-ABL1 values across a measuring range of more than six orders of magnitude. For the given data set,

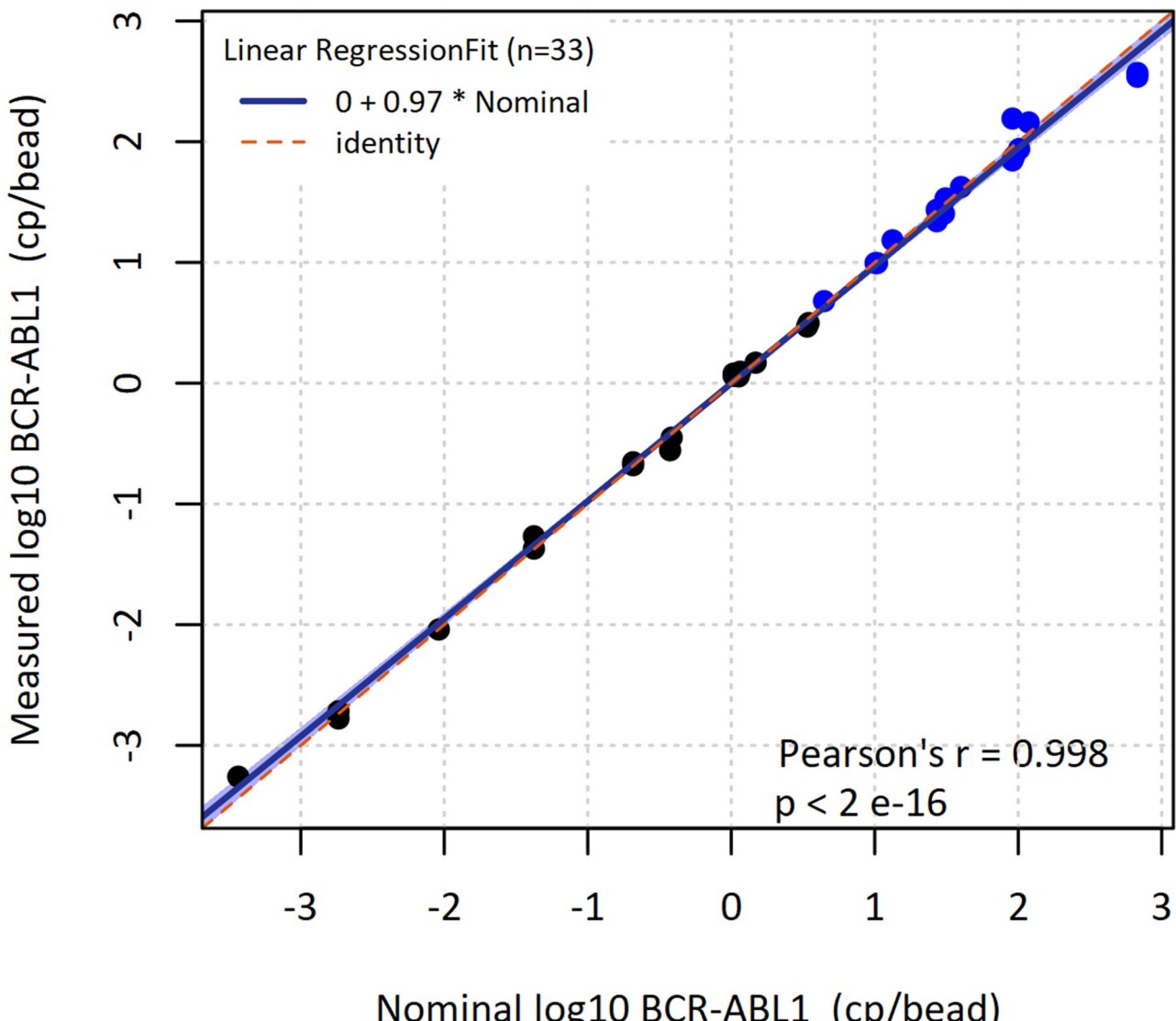

**Fig 6. BCR-ABL1 measuring range.** Input copies per bead are plotted on the x-axis against the measured copies per bead on the y-axis (both logarithmic scale; black, Poisson analysis of digital PCR data; blue, Ct-value analysis of quantitative real-time PCR).

real-time quantitative PCR extends the measuring range of digital PCR by more than two orders of magnitude.

We assessed the repeatability of the assay with three contrived samples comprising different ratios BCR-ABL1/GUSB (low, medium, high). The chosen BCR-ABL1/GUSB levels reflect the clinically relevant range to monitor the disease status in the context of CML therapy [22, 23]. Each sample was analyzed with six replicates. Replicates were tested in different laboratory runs, as the nanoreactor beads were configured to process one sample at a time. Therefore, repeatability conditions include between-run imprecision. Results for the precision of quantification for BCR-ABL1 (absolute) and GUSB (absolute) are summarized in Table 2.

The precision results for ratios of BCR-ABL1/GUSB are shown in Table 3.

The concentration of GUSB target was set to a mean value of 2.05 log10 cp/bead for all levels resulting in approximately 1,100,000 GUSB copies in 10,000 beads. The lowest measured

**Table 2. Precision results for BCR-ABL1 and GUSB.**

| Level | Replicates | Target | Mean [log10 cp/bead] | Mean* [cp/reaction] | SD [log10 cp/bead] | Analysis Method |
|---|---|---|---|---|---|---|
| Low | 6 | BCR-ABL1 | -2.72 | 19 | 0.09 | Poisson |
| | | GUSB | 2.04 | 1,100,000 | 0.02 | Ct |
| Medium | 6 | BCR-ABL1 | -1.03 | 933 | 0.04 | Poisson |
| | | GUSB | 2.00 | 1,000,000 | 0.04 | Ct |
| High | 6 | BCR-ABL1 | 1.85 | 708,000 | 0.04 | Ct |
| | | GUSB | 2.10 | 1,260,000 | 0.02 | Ct |

* for 10,000 beads per reaction; SD, standard deviation.

ratio -2.76 log10% corresponds to a molecular response between 4.5 and 5. $MR^{4.5}$ and $MR^5$ is considered state of the art for the detection and quantification limit of the method for clinical applications [21, 23]. To ensure assay sensitivity required to achieve $MR^5$, a minimum number of 240,000 copies of control gene transcript GUSB is recommended [22]. Because of the high capacity of the beads our assay is exceeding this recommendation by a factor of four.

In comparison, the current technically leading commercially available test features a precision of 0.25 log10 for $MR^{\leq 4.6}$ (QXDx™ BCR-ABL1%IS Kit by Biorad digital PCR) [21]. This precision claim takes all variance components into account, including different instruments, reagent lots, operators, and analytical repeatability. However, repeatability is by far the strongest contributor to variance. Therefore, the achieved repeatability with the nanoreactor beads of 0.09 log10% for MR between 4.5 and 5 can be considered excellent. The implementation of real-time quantitative PCR analysis for nanoreactor beads provides favorable precision results because variance from the signal detection process is strongly reduced by averaging Ct-values from approximately 550 independent nanoreactor beads.

Peripheral blood samples from CML patients were used to compare test results obtained with the standard method based on cDNA synthesis followed by real time PCR against the new assay format with the nanoreactor beads. This tests were performed with approximately 10,000 beads per run. With an allowed maximum of 5.3 targets per bead, the upper limit of quantification for the digital readout is 53,000 targets per test. For quantification of higher target numbers real-time quantitative PCR analysis based on the established calibration curves has been applied. A total of 28 clinical specimens from CML patients were provided by the Department of Hematology at JUH in Germany. The ratios BCR-ABL1/GUSB of the reference method ranged from -3.52 log10% (0.0003 BCR-ABL1/GUSB%) to 1.36 log10% (23 BCR-ABL1/GUSB%). Among the 28 patient samples 67.8% of the cases (19/28) were shown to be detectable and quantified by the new method and the reference method, while 10.7% of cases (3/28) were shown to be negative by both tests. 14.3% of cases (4/28) were detected by

**Table 3. Precision results for ratios BCR-ABL1/GUSB.**

| Level | Ratio BCR-ABL1/GUSB | | | | Analysis Method for BCR-ABL1/ GUSB |
|---|---|---|---|---|---|
| | Mean [log10%] | Mean [%] | Mean MR | SD [log10%] | |
| Low | -2.76 | 0.0017 | 4.76 | 0.09 | Poisson/Ct |
| Medium | -1.04 | 0.091 | 3.04 | 0.01 | Poisson/Ct |
| High | 1.75 | 56 | 0.25 | 0.06 | Ct/Ct |

SD, standard deviation; MR, molecular response; Ct, threshold cycle.

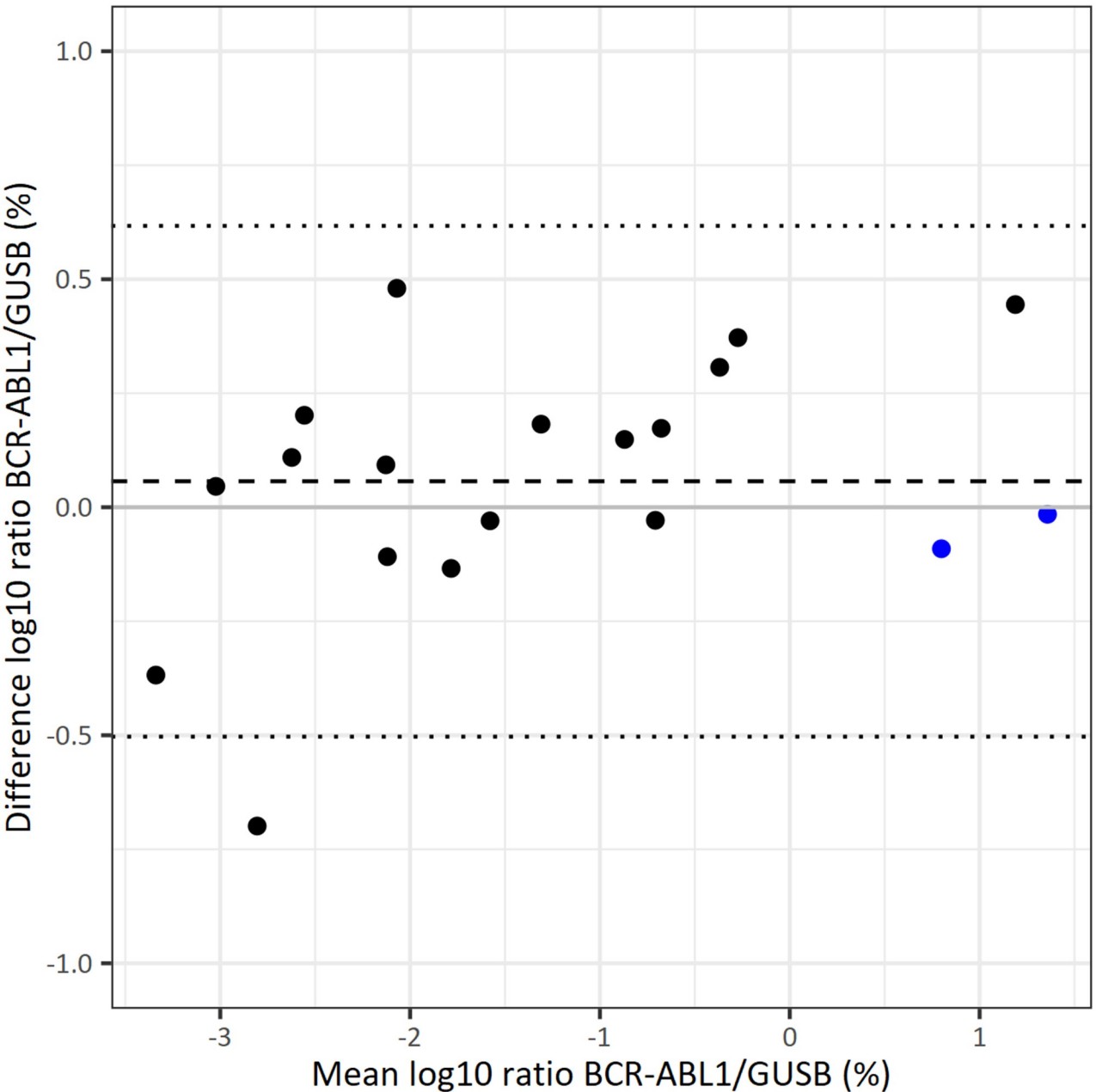

**Fig 7. Bland Altman plot for data obtained with the reference and the new method.** Black, Poisson analysis of digital PCR data for BCR-ABL1; blue, Ct-value analysis of quantitative real-time PCR for BCR-ABL1).

reference method and undetected by the new method. 7.1% of cases (2/28) were undetected by the reference method and detected by the new method.

A Bland Altman plot analysis was performed using the 19 quantitative results. The Bland Altman plot in Fig 7 shows a small constant bias between both methods across the complete measuring range of 0.06 log10%. This bias can be eliminated with a conversion factor. The figure also shows the upper and lower 2SD of the mean difference that was observed with SD = 0.28 log10%. There is no indication that the variation is dependent on the BCR-ABL1 transcript level.

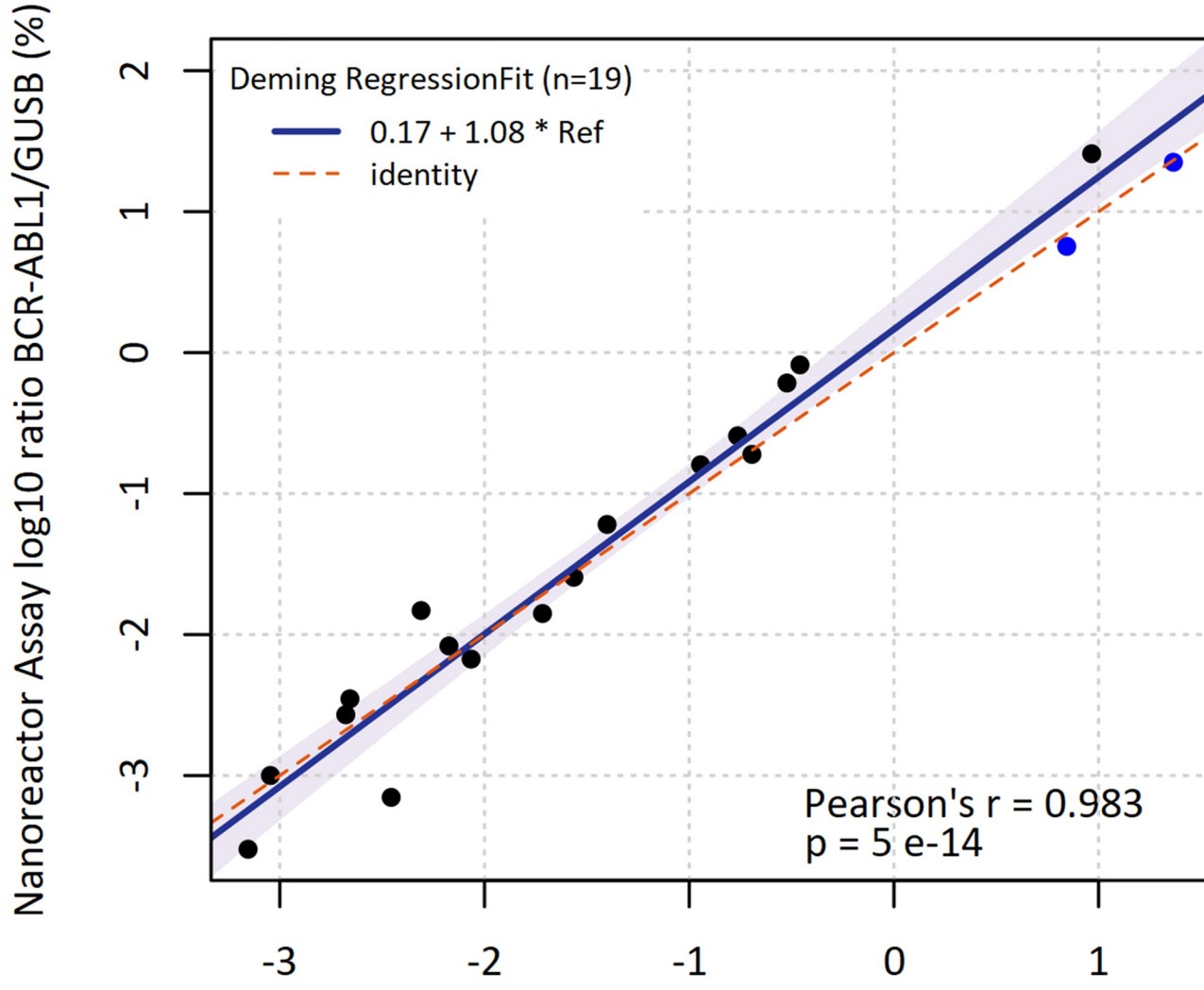

**Fig 8. Deming regression for data obtained with the reference and the new method.** Black, Poisson analysis of digital PCR data for BCR-ABL1; blue, Ct-value analysis of quantitative real-time PCR for BCR-ABL1, Ref = reference).

Fig 8 shows the Deming regression fit for the 19 quantitative results obtained both with the standard laboratory test assay (cDNA synthesis, real time PCR) and with nanoreactor beads. The slope was determined with 1.08 with 95% CI from 0.99 to 1.21. The intercept was 0.17 with 95% CI from 0.03 to 0.38. Excellent concordance between the nanoreactor bead assay and the reference with a strong linear relationship (Pearsons' r = 0.983, p = $5 \times 10^{-14}$) has been observed.

## Conclusions

Nanoreactor beads are a new class of molecular reagents enabling microfluidics free digital molecular assays. The theoretical upper limit of quantification based on Poisson analysis corresponds to approximately 10x the number of nanoreactor beads employed in the assay. To

overcome this limitation, we supplemented the analysis of digital PCR with real-time quantitative PCR analysis for target concentrations exceeding the upper limit of digital quantification. This approach enabled us to design an assay for the quantification of BCR-ABL1, GUSB, and for the ratios of BCR-ABL1/GUSB with a dynamic range of >6 orders of magnitude and to test the assay on clinical peripheral blood samples. The results indicate that the developed approach provides for high sensitivity and, to our knowledge, an unprecedented measurement range including digital PCR in a single reaction. Moreover, the assay is simple to perform and delivers quantitative results comparable with the current clinical laboratory standard method.

## Supporting information

**S1 Fig. FAM and Cy5 Fluorescence of nanoreacctor beads at 62°C and 20°C.** Shown images have been acquired after a PCR run with 40 cycles; FAM, fluorescence of GUSB gene probe; Cy5, fluorescence of BCR-ABL1 gene probe; bright and dark fluorescence represents PCR positive and negative nanoreactor beads, respectively. Positive and negative nanoreactor beads are distinguishable at 62°C and at 20°C.
(TIF)

**S2 Fig. Reaction and detection chamber (RDC).** Right, Dimensions of RDC are shown at the right in mm; 0.1 indicates the distance between the transparent cover and the thin sheet. Left, a cross section of a rendered drawing of the RDC. Enlarged inset shows set-up forming a flat chamber for accommodating nanoreactor beads.
(TIF)

**S1 Table. Patient sample data.**
(DOCX)

## Acknowledgments

We wish to thank the patients who have agreed to provide the samples for our method comparison. For excellent technical work, we like to thank Ines Engelman (BLINK AG, Jena, Germany) and Anja Waldau (Universitätsklinikum Jena, Klinik für Innere Medizin II, Abteilung Hämatologie und Internistische Onkologie, Jena, Germany).

## Author Contributions

**Conceptualization:** Ivan Francisco Loncarevic, Susanne Toepfer, Thomas Ellinger, Thomas Ernst, Andreas Hochhaus, Eugen Ermantraut.

**Funding acquisition:** Andreas Hochhaus, Eugen Ermantraut.

**Investigation:** Ivan Francisco Loncarevic, Stephan Hubold, Susanne Klingner, Lea Kanitz, Thomas Ellinger, Katrin Steinmetzer.

**Methodology:** Stephan Hubold, Lea Kanitz.

**Project administration:** Eugen Ermantraut.

**Supervision:** Ivan Francisco Loncarevic, Eugen Ermantraut.

**Validation:** Susanne Toepfer, Stephan Hubold.

**Writing – original draft:** Susanne Toepfer, Eugen Ermantraut.

**Writing – review & editing:** Ivan Francisco Loncarevic, Susanne Toepfer, Thomas Ernst, Andreas Hochhaus.

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
