## [Decision Letter · Decision Letter 0]

23 Nov 2020

PONE-D-20-33329

Ultra-precise quantification of mRNA targets across a broad dynamic range with nanoreactor beads

PLOS ONE

Dear Dr. Loncarevic,

Thank you for submitting your manuscript to PLOS ONE. After careful consideration, we feel that it has merit but does not fully meet PLOS ONE’s publication criteria as it currently stands. Therefore, we invite you to submit a revised version of the manuscript that addresses the points raised during the review process.

We look forward to receiving your revised manuscript.

Kind regards,

Ruslan Kalendar, PhD

Academic Editor

PLOS ONE

Journal Requirements:

We note that one or more of the authors are employed by a commercial company: BLINK AG.

2.1. Please provide an amended Funding Statement declaring this commercial affiliation, as well as a statement regarding the Role of Funders in your study. If the funding organization did not play a role in the study design, data collection and analysis, decision to publish, or preparation of the manuscript and only provided financial support in the form of authors' salaries and/or research materials, please review your statements relating to the author contributions, and ensure you have specifically and accurately indicated the role(s) that these authors had in your study. You can update author roles in the Author Contributions section of the online submission form.

2.2. Please also provide an updated Competing Interests Statement declaring this commercial affiliation along with any other relevant declarations relating to employment, consultancy, patents, products in development, or marketed products, etc.  

 3. In your Methods section, please provide additional information about the participant recruitment method and the demographic details of your participants. Please ensure you have provided sufficient details to replicate the analyses such as: a) the recruitment date range (month and year), b) a description of any inclusion/exclusion criteria that were applied to participant recruitment, c) a table of relevant demographic details, d) a statement as to whether your sample can be considered representative of a larger population, e) a description of how participants were recruited, and f) descriptions of where participants were recruited and where the research took place.

5. We note you have included a table to which you do not refer in the text of your manuscript. Please ensure that you refer to Table 2 in your text; if accepted, production will need this reference to link the reader to the Table.

Reviewers' comments:

Reviewer's Responses to Questions

**Comments to the Author**

1. Is the manuscript technically sound, and do the data support the conclusions?

Reviewer #1: Yes

Reviewer #2: Yes

Reviewer #3: Yes

2. Has the statistical analysis been performed appropriately and rigorously? 

Reviewer #1: Yes

Reviewer #2: Yes

Reviewer #3: I Don't Know

3. Have the authors made all data underlying the findings in their manuscript fully available?

Reviewer #1: Yes

Reviewer #2: Yes

Reviewer #3: Yes

4. Is the manuscript presented in an intelligible fashion and written in standard English?

Reviewer #1: Yes

Reviewer #2: Yes

Reviewer #3: Yes

5. Review Comments to the Author

Reviewer #1: Overall, the manuscript is well written. The following are some suggestions for the authors to consider for a revised manuscript.

Major:

Since this is mainly an assay development manuscript, the data on assay quality assessment are somewhat insufficient. Only the linear dynamic range is provided. What about assay specificity? Intra-day vs. inter-day reproducibility? Is there any optimization data on beads: samples ration? How about the LLOQ of this assay?

Minor:

All the texts in every figures are fuzzy and cannot to be recognized.

Reviewer #2: The authors present a way to compartmentalize biomolecules using hydrogel beads. They explain very well why this is important in the introduction to achieve true quantitative measurements of nucleic acids as current systems having the q in their name are not really quantitative. This is a very important issue as the COVID19 pandemia have demonstrated. The manuscript is technically sound and well structured. I must say I enjoyed reviewing this manuscript. Hence, I do recommend its publication with some minor revisions. Below, there are a number of recommendations and questions that I'd like the authors to address.

I suggest authors to add a sentence in the introduction to describe the differences between their method and the ones in ref 7 and 8 of their manuscript as this is an important justification of their work.

Regarding the cell based assays authors should refer how cells were cultured and also number of cells used. These number of cells could be used to translate to the number of transcript per cell detected. The number of a transcript per cell is always difficult to determine hence this technology could be useful to do so.

One aspect which is not clear to me is the melting and reforming step of the process. In Fig. 1 (iv.) is said "bead melting, PCR and fluorescence detection". As the reading is done with beads, how long does it take to reform the beads? Are beads melted during the whole PCR process? So, I suggest to explain this step with more detail and add pictures of the 3 min denaturation step either in the main text or supplementary information.

When a sample gives #dark beads <0.5%, it is then quantified by Ct and calibration curve. If that sample is diluted to have a #dark beads >0.5%, it is quantified by Poisson analysis. Are the quantifications equal, taking into account the dilution factor? Fig. 5 might answer it but it is not clear to me if that samples are serial dilutions or not.

Do you need to do a calibration curve per run?

Duration of each step in the workflow scheme could be useful for the reader to understand how long the sample-to-result takes.

More details of the commercial suppliers of reagents are needed. It might be good to have a General section at the beginning of the Material and Methods section containing these details.

TaqMan probes references are not provided.

While authors give a reference for primers and probes, a Table describing them would be very helpful for readers

Fig. 1. A scale bar is missing

Line 247 there is a reference error

Reviewer #3: The authors report the development of a 'naoreactor bead' based assay for the quantification of mRNA targets. It appears that the use of nanoreactor beads can be used in quantification of mRNA targets spanning six order of magnitude.

The authors build their manuscript on an argument by stating that digital assays are complex and require sophisticated microfluidic tools. Unless I am missing some important aspects/details the authors do generate 'nanoreactor beads' using a microfluidics chip, and the proposed method by all means not an unsophisticated one. Hence it is not clear to me why they prefer to 'sell' their method to be microfluidics-free and 'easy'.

The manuscript text appears to be not submission ready given the overwhelming number of points I could comment on. The text could benefit from a thorough editing, especially paying attention to inconsistent terminology use that confused me on multiple occasions, especially the nanoreactor, digital nanoreactors, and alike…

There is also no statistical analysis section provided within the Materials and Methods section. And finally the method section can benefit from a detail oriented focus.

Given the current status of the manuscript, I would conclude resubmission after major revision; however, in this journal it corresponds to major revision hence my decision.

Below I list my various comments based on the page and line numbers:

Cover page: there are typos in the keywords!

Page 2, Line 16: good agreement -> Can the authors use a quantitative measure. Also while the title reads 'ultra-precise' the closing statement of the abstract states 'good agreement'. It is ultra-precise or good? (I personally dislike such use of prepositions. [What would be the next stage? Ultra²?])

P3, L23: partitions -> It is not clear what is meant by partition. Can the authors clarify (the figure 1 legend reads beads. Can you please tune the term use for consistency/clarity).

P3, L24: Poisson criteria -> Poisson distribution (criteria?). Can you please clarify what the criteria is/are?

P3, L29: digital compartments -> for consistent terminology use can you please stick to one term whether it is the digital compartments/beads/nanoreactor compartment/nanoreactor beads/microbeads/sub-nanoliter droplets (whichever term the authors prefer so long as it is used consistently throughout the manuscript.)

P3, L29: generating digital compartments -> Can you present how your droplet generation does not necessitate microfluidics as compared to the standard. Also lay out the details of the concrete differences between the two techniques for droplet generation.

P3, L35: 'amplification reagents and the extracted nucleic acid.' -> 'RT-PCR amplification reagents and the extracted nucleic acids (GUS8 and BCR-ABL RNA).'

Also the text reads BCR-ABL1 RNA. Please stick to one term.

P3, L 40: Please state the size of the beads.

Figure 1. In panel a please add sub-sections i., ii, … so that the figure complies with the figure legend.

Figure 1b and c. Can you provide the scale.

P4, L47: extensive shaking -> Can you provide reproducible quantitative information.

Figure 2 legend:

. Can you please use the same terms both in the figure and the legend, i.e. Fluorescence Micro-imager vs epi-fluorescence microscope, BLINK vs Blink…

. Can you provide the technical details of the chamber i.e. material, dimensions, thickness, alike… in the legend.

. Please state the company and also provide a link to the software/website.

P5, L69: nanoreactor compartment -> see comment P3, L29

P5, L70: modest number -> Please provide quantitative information wherever possible.

P5, L71: highly precise -> Can the authors avoid the use of qualitative adverb and state the quantitative findings.

P5, L89: fluidic connection between tubing and chip -> Please provide a link to the chip or provide the design in supplementary.

P5, L99: was removed -> was removed by?

P6, L112: e13a2 or e14a2 BCR-ABL1 -> Please describe what e13a2 and e14a2 are.

P7, L113: BCR-ABL-1 -> BCR-ABL1

P7, L119: HL60 and K562 -> Please describe what are HL60 and K562.

P7, L124: as described elsewhere -> Please provide all the necessary details in the methods section.

P8, L143: latter -> There is a single removal step in the previous sentence. Please re-phrase.

P8, L148: parameters -> Please provide the full details.

Figure 3 legend: Please describe the different beads in the legend. Preferably arrows can also be used to highlighting the specific beads.

P11, L208: ERM-AD623 -> What is ERM-AD623? Please describe.

P11, L209: (data not shown) -> Please provide the necessary data.

P11, L211: high copy numbers -> high copy numbers (of what?)

P11, L213: The concentration levels represent the expected measuring range for clin(ic)al samples -> Please provide data or a reference to support this statement.

Figure 5: BCRLAB -> BCR-ABL1.

Also please use different colours for dots and triangles.

P13, L247: Table 1. Error! Reference source not found. -> Cross reference error?

P14, L260; L267: The molecular response (MR) -> Acronym was already introduced in line 258.

P14, L267: considered -> Please provide a reference for this statement.

P14, L273: precision -> Please provide a reference for this statement.

P15, L285: 53.000 -> 53,000?

P15, L294: Blink test -> What is a Blink test?

P16, L303: the reference method -> Please help the reader by providing a legend that provides all the relevant information.

P16, L307: same data -> which data?

P16, L318: microfluidics free digital molecular assays -> Unless I am missing the picture, the nanoreactor beads were generated using a Dolomite microfluidics chip. So how can the authors claim that the process is microfluidics free?

P16, L320: bead-nanoreactors -> is this the same as nanoreactor beads?

P16, L324: for the ratios -> for the ratios of

P16, L329: current laboratory standard -> which standard?

Figure 6 and 7: BCRLAB -> BCR-ABL1.

(applies to the entire text)

. real time - real-time

. rtPCR - RT-PCR

. have a white space before units, e.g. #µL -> # µL.

. em dash between numerical values, e.g. 4-8°C -> 4–8°C

. use a minus sign and not a dash sign.

6. PLOS authors have the option to publish the peer review history of their article (what does this mean?). If published, this will include your full peer review and any attached files.

Reviewer #1: No

Reviewer #2: **Yes: **Juan J. Diaz-Mochon

Reviewer #3: **Yes: **Rahmi Lale

---

## [Author Response · Author response to Decision Letter 0]

13 Jan 2021

Response to Reviewer 1 

Overall, the manuscript is well written. The following are some suggestions for the authors to consider for a revised manuscript.

Major:

Since this is mainly an assay development manuscript, the data on assay quality assessment are somewhat insufficient. Only the linear dynamic range is provided. What about assay specificity? Intra-day vs. inter-day reproducibility? Is there any optimization data on beads: samples ration? How about the LLOQ of this assay?

• In this manuscript we are demonstrating the feasibility of a novel approach for setting up molecular assays with broad dynamic range and high precision based on combined digital and real time readout of fluorescence generated within nanoreactors beads. Since we are presenting basic technical feasibility the clinical application is to be considered only an exemplary model for the kind of assays that would benefit from the new technique. Of course, for use in a clinical context further validation is required.

Therefore for this initial feasibility assessment we would like to limit the scope of assay characterization to a qualitative assessment of the achieved precision shown by the intraday SD with 6 replicates. The measuring range and linearity by the comparison to the reference laboratory data provides an initial data set for achievable specificity with this assay format. 

Minor:

All the texts in every figures are fuzzy and cannot to be recognized. 

• All images have now been set to 300dpi. 

Response to Reviewer 2 

The authors present a way to compartmentalize biomolecules using hydrogel beads. They explain very well why this is important in the introduction to achieve true quantitative measurements of nucleic acids as current systems having the q in their name are not really quantitative. This is a very important issue as the COVID19 pandemia have demonstrated. The manuscript is technically sound and well structured. I must say I enjoyed reviewing this manuscript. Hence, I do recommend its publication with some minor revisions. Below, there are a number of recommendations and questions that I'd like the authors to address.

I suggest authors to add a sentence in the introduction to describe the differences between their method and the ones in ref 7 and 8 of their manuscript as this is an important justification of their work. 

• A sentence is now added in line 31- 34 clearly outlining the difference between the referenced technologies and the new method presented in this manuscript. 

Regarding the cell-based assays authors should refer how cells were cultured and also number of cells used. These number of cells could be used to translate to the number of transcript per cell detected. The number of a transcript per cell is always difficult to determine hence this technology could be useful to do so. 

• HL60 cells and K562 cells served only as source for extraction of RNA negative and positive for BCR-ABL1, respectively. Both RNA extracts where used to generate contrived samples with low, middle and high BCR-ABL1/GUSB ratios. Though the presented method indeed could be applied to quantitate transcripts per cell, in this work the actual recovery of RNA-copies per cell did not play any role. We have now clarified this in the text.

One aspect which is not clear to me is the melting and reforming step of the process. In Fig. 1 (iv.) is said "bead melting, PCR and fluorescence detection". As the reading is done with beads, how long does it take to reform the beads? Are beads melted during the whole PCR process? So, I suggest to explain this step with more detail and add pictures of the 3 min denaturation step either in the main text or supplementary information. 

• This is now better explained in the text (line 42-71)

When a sample gives #dark beads <0.5%, it is then quantified by Ct and calibration curve. If that sample is diluted to have a #dark beads >0.5%, it is quantified by Poisson analysis. Are the quantifications equal, taking into account the dilution factor? Fig. 5 might answer it but it is not clear to me if that samples are serial dilutions or not.

Do you need to do a calibration curve per run? 

• A calibration curve is required for the real-time analysis part of the assay, this is if less than 0.5% present in the imaged area remain dark. If the number is >0.5% absolute quantification based on Poisson analysis is feasible. We have added a clarifying statement to the data analysis section.

• Adjustment of the low, middle, and high level of BCR-ABL1/ratio is not done by serial dilution. We have mixed extracted RNA from different cell lines at different ratios to order to obtain three distinct levels of BCR-ABL1/GUSB ratios within 6 orders of magnitude.

Duration of each step in the workflow scheme could be useful for the reader to understand how long the sample-to-result takes.

• The duration of the test from sample to result is added (L188-L190)

More details of the commercial suppliers of reagents are needed. It might be good to have a General section at the beginning of the Material and Methods section containing these details.

TaqMan probes references are not provided.

While authors give a reference for primers and probes, a Table describing them would be very helpful for readers

• A table including all sequences for primers and probes has been added to the text. Wherever appropriate additional information on the source for the used materials has been provided.

Fig. 1. A scale bar is missing 

• added

Line 247 there is a reference error 

• corrected

Response to Reviewer 3 

The authors report the development of a 'nanoreactor bead' based assay for the quantification of mRNA targets. It appears that the use of nanoreactor beads can be used in quantification of mRNA targets spanning six order of magnitude.

The authors build their manuscript on an argument by stating that digital assays are complex and require sophisticated microfluidic tools. Unless I am missing some important aspects/details the authors do generate 'nanoreactor beads' using a microfluidics chip, and the proposed method by all means not an unsophisticated one. Hence it is not clear to me why they prefer to 'sell' their method to be microfluidics-free and 'easy'. 

• The presented assay format is based on the use of hydrogel beads. These beads are made in bulk in large quantities and stored until they are needed. Contrary conventional droplet PCR requires microfluidic tools and specialized equipment to generate mono-disperse droplet nano-compartments to run the actual test, whereas in the presented format the beads in combination with a fluorocarbon oil and a suitable emulsifier are sufficient for partitioning of the sample into sub-nanoliter sized digital reaction compartments.

We have now made this clear in the text.

The manuscript text appears to be not submission ready given the overwhelming number of points I could comment on. The text could benefit from a thorough editing, especially paying attention to inconsistent terminology use that confused me on multiple occasions, especially the nanoreactor, digital nanoreactors, and alike… - 

• Thank you for this input: We have checked the text for consistency with regards to the terminology - 

- nanoreactor beads is now the only term used for the beads

- Reaction and detection chamber us used throughout the text

There is also no statistical analysis section provided within the Materials and Methods section. 

• A statistical analysis section has been added to the Materials and Methods

And finally the method section can benefit from a detail oriented focus.

Given the current status of the manuscript, I would conclude resubmission after major revision; however, in this journal it corresponds to major revision hence my decision. Below I list my various comments based on the page and line numbers:

Cover page: there are typos in the keywords! 

• Corrected

Page 2, Line 16: good agreement -> Can the authors use a quantitative measure. Also while the title reads 'ultra-precise' the closing statement of the abstract states 'good agreement'. It is ultra-precise or good? (I personally dislike such use of prepositions. [What would be the next stage? Ultra²?]) 

• Level of concordance has been quantified by Pearsons correlation coefficient of 0.983 and slope of 1.08.

P3, L23: partitions -> It is not clear what is meant by partition. Can the authors clarify (the figure 1 legend reads beads. Can you please tune the term use for consistency/clarity). 

• We have added a description what partition means in line 23. 

In this manuscript we are referring to the basic concepts of digital PCR whereof sample partitioning is the fundamental approach. The sample is divided into a number of identical sub-volumes. A partition represents such a unit sub-volume. Since Fig 1 is referring to the newly developed process the figure capture refers to “nanoreactor beads”, whereas the introductory sentences are referring to digital analysis in general, hence our use of “partitions” there.

P3, L24: Poisson criteria -> Poisson distribution (criteria?). Can you please clarify what the criteria is/are? 

• The unclear statement has been removed. The underlying mathematical model is explained in the quoted reference [5].

P3, L29: digital compartments -> for consistent terminology use can you please stick to one term whether it is the digital compartments/beads/nanoreactor compartment/nanoreactor beads/microbeads/sub-nanoliter droplets (whichever term the authors prefer so long as it is used consistently throughout the manuscript.)

• Thank you, this has been addressed 

P3, L29: generating digital compartments -> Can you present how your droplet generation does not necessitate microfluidics as compared to the standard. Also lay out the details of the concrete differences between the two techniques for droplet generation. 

• A clarifying sentence has been added (P3, L31-34)

P3, L35: 'amplification reagents and the extracted nucleic acid.' -> 'RT-PCR amplification reagents and the extracted nucleic acids (GUS8 and BCR-ABL RNA).'

Also the text reads BCR-ABL1 RNA. Please stick to one term. 

• We have checked the text for consistency with regards to the terminology.

P3, L 40: Please state the size of the beads.

• Scale bar and information on size and volume of the beads has been added to the figure caption 

Figure 1. In panel a please add sub-sections i., ii, … so that the figure complies with the figure legend.

• The changes have been made to Fig 1

Figure 1b and c. Can you provide the scale.

• Scale bars have been added to the image

P4, L47: extensive shaking -> Can you provide reproducible quantitative information.

• The process is now described in more detail in line 167-171 in the materials and methods section. 

Figure 2 legend:

Can you please use the same terms both in the figure and the legend, i.e. Fluorescence Micro-imager vs epi-fluorescence microscope, BLINK vs Blink… the text was checked for consistency

. Can you provide the technical details of the chamber i.e. material, dimensions, thickness, alike… in the legend. Please state the company and also provide a link to the software/website.

• Information has been added to figure legend

P5, L69: nanoreactor compartment -> see comment P3, L29

• The sentence has been modified for clarity

P5, L70: modest number -> Please provide quantitative information wherever possible.

• The quantity has been added.

P5, L71: highly precise -> Can the authors avoid the use of qualitative adverb and state the quantitative findings.

• We have modified the sentence.

P5, L89: fluidic connection between tubing and chip -> Please provide a link to the chip or provide the design in supplementary. 

• the described set up used for generating nanoreactor beads represents a variant of the commercially available µEncapsulator system. The setup and components are described in detail in the product literature of Dolomite microfluidic. The manufacturer is referenced accordingly.

P5, L99: was removed -> was removed by? 

• Sentence has been corrected

P6, L112: e13a2 or e14a2 BCR-ABL1 -> Please describe what e13a2 and e14a2 are. 

• we have changed the names of the transcripts to the more common and known nomenclature (b2a2 and b3a2) and added a short description.

These are transcript variants that are typical for the CML phenotype. A possible differential impact of the respective transcript variants e13a2 ("b2a2") and e14a2 ("b3a2") on disease phenotype and outcome is still a subject of debate. 

• We confined the analysis to samples with the common genotypes in order to focus on the technology rather than the disease. Thus, we also added a statement to the sample selection and sample exclusion strategy in section samples.

P7, L113: BCR-ABL-1 -> BCR-ABL1 OK

• Now consistent BCR-ABL1.

P7, L119: HL60 and K562 -> Please describe what are HL60 and K562. 

• A description and references are added (L166-L167).

P7, L124: as described elsewhere -> Please provide all the necessary details in the methods section. 

• A table is added including all sequences for primers and probes 

P8, L143: latter -> There is a single removal step in the previous sentence. Please re-phrase. 

• OK

P8, L148: parameters -> Please provide the full details.

• all necessary information is provided in the subsequent sentence.

Figure 3 legend: Please describe the different beads in the legend. Preferably arrows can also be used to highlighting the specific beads. 

• Arrows have been added to the images for better understanding and the legend modified, accordingly 

P11, L208: ERM-AD623 -> What is ERM-AD623? Please describe. 

• ERM-AD623 is described in L172-175 

P11, L209: (data not shown) -> Please provide the necessary data.

• Removed, as not necessary in this context

P11, L211: high copy numbers -> high copy numbers (of what?)

• Sentence has been corrected

P11, L213: The concentration levels represent the expected measuring range for clin(ic)al samples -> Please provide data or a reference to support this statement. 

• References supporting the statement have been included

Figure 5: BCRLAB -> BCR-ABL1. 

Also please use different colours for dots and triangles.

• Images have been modified

P13, L247: Table 1. Error! Reference source not found. -> Cross reference error?

• Corrected

P14, L260; L267: The molecular response (MR) -> Acronym was already introduced in line 258.

• Corrected

P14, L267: considered -> Please provide a reference for this statement.

• References supporting the statement have been included

P14, L273: precision -> Please provide a reference for this statement. 

• References supporting the statement have been included

P15, L285: 53.000 -> 53,000?

• Corrected

P15, L294: Blink test -> What is a Blink test? 

• Replaced with “new test”

P16, L303: the reference method -> Please help the reader by providing a legend that provides all the relevant information.

• Reference method has been explained in line 139 - 140

P16, L307: same data -> which data? Fig. 7 

• Rephrased and information competed (Fig 8., L512)

P16, L318: microfluidics free digital molecular assays -> Unless I am missing the picture, the nanoreactor beads were generated using a Dolomite microfluidics chip. So how can the authors claim that the process is microfluidics free? 

• Our statement is correct, as no specific microfluidic tools are required to split the sample into digital partitions during the test, instead we use pre-made hydrogel beads off the shelf for partitioning the targets into digital compartments.

P16, L320: bead-nanoreactors -> is this the same as nanoreactor beads?

• Corrected for consistency

P16, L324: for the ratios -> for the ratios of 

• corrected

P16, L329: current laboratory standard -> which standard? 

• Rephrased for clarity

---

## [Decision Letter · Decision Letter 1]

5 Feb 2021

Ultra-precise quantification of mRNA targets across a broad dynamic range with nanoreactor beads

PONE-D-20-33329R1

Dear Dr. Loncarevic,

We’re pleased to inform you that your manuscript has been judged scientifically suitable for publication and will be formally accepted for publication once it meets all outstanding technical requirements.

Kind regards,

Ruslan Kalendar, PhD

Academic Editor

PLOS ONE

Reviewers' comments:

Reviewer's Responses to Questions

**Comments to the Author**

Reviewer #1: All comments have been addressed

Reviewer #3: All comments have been addressed

2. Is the manuscript technically sound, and do the data support the conclusions?

Reviewer #1: Yes

Reviewer #3: Yes

3. Has the statistical analysis been performed appropriately and rigorously? 

Reviewer #1: Yes

Reviewer #3: Yes

4. Have the authors made all data underlying the findings in their manuscript fully available?

Reviewer #1: Yes

Reviewer #3: Yes

5. Is the manuscript presented in an intelligible fashion and written in standard English?

Reviewer #1: Yes

Reviewer #3: Yes

6. Review Comments to the Author

Reviewer #1: (No Response)

Reviewer #3: I would like to thank authors for implementing the changes that improved the manuscript.

There is still a (cross-reference) error in line 344.

7. PLOS authors have the option to publish the peer review history of their article (what does this mean?). If published, this will include your full peer review and any attached files.

Reviewer #1: **Yes: **Tai-Du Lin

Reviewer #3: **Yes: **Rahmi Lale

---

## [Editor Report · Acceptance letter]

8 Mar 2021

PONE-D-20-33329R1 

Ultra-precise quantification of mRNA targets across a broad dynamic range with nanoreactor beads 

Dear Dr. Loncarevic:

I'm pleased to inform you that your manuscript has been deemed suitable for publication in PLOS ONE. Congratulations! Your manuscript is now with our production department. 

Kind regards, 

on behalf of

Prof. Ruslan Kalendar 

Academic Editor

PLOS ONE